# Multi-Faceted Role of Luteolin in Cancer Metastasis: EMT, Angiogenesis, ECM Degradation and Apoptosis

**DOI:** 10.3390/ijms24108824

**Published:** 2023-05-16

**Authors:** Maria Teresa Rocchetti, Francesco Bellanti, Mariia Zadorozhna, Daniela Fiocco, Domenica Mangieri

**Affiliations:** 1Department of Clinical and Experimental Medicine, University of Foggia, Via Pinto 1, 71122 Foggia, Italy; mariateresa.rocchetti@unifg.it (M.T.R.); daniela.fiocco@unifg.it (D.F.); 2Department of Medical and Surgical Sciences, University of Foggia, Via Pinto 1, 71122 Foggia, Italy; francesco.bellanti@unifg.it; 3Medical Genetics Unit, Department of Molecular Medicine, University of Pavia, Via Forlanini 14, 27100 Pavia, Italy; mariia.zadorozhna@unipv.it

**Keywords:** phytochemicals, flavonoids, metastasis prevention, signaling pathways

## Abstract

Luteolin (3′,4′,5,7-tetrahydroxyflavone), a member of the flavonoid family derived from plants and fruits, shows a wide range of biomedical applications. In fact, due to its anti-inflammatory, antioxidant and immunomodulatory activities, Asian medicine has been using luteolin for centuries to treat several human diseases, including arthritis, rheumatism, hypertension, neurodegenerative disorders and various infections. Of note, luteolin displays many anti-cancer/anti-metastatic properties. Thus, the purpose of this review consists in highlighting the relevant mechanisms by which luteolin inhibits tumor progression in metastasis, i.e., affecting epithelial-mesenchymal transition (EMT), repressing angiogenesis and lysis of extracellular matrix (ECM), as well as inducing apoptosis.

## 1. Introduction

Cancer is a complex disease, representing the second leading cause of death in the world [1]. In fact, despite the advances in diagnostic and therapeutic protocols—which include surgery, radiotherapy and chemotherapy, as well as target and gene therapy—the survival rate for patients suffering of cancer remains very poor [2,3,4,5,6]. Probably, the real problem of high cancer mortality is its relapse after months to years, through the occurrence of metastases, an event that consists in the spread of cancer cells from a primary lesion to distant sites [7,8,9]. Currently available cancer/metastatic treatments often induce high toxicity and are associated with several types of adverse events, which are frequently unpredictable and unexplained [10,11,12,13,14,15]. Thus, finding novel and more efficacious strategies to treat cancer and prevent metastasis is strongly encouraged [8,16,17,18,19]. Because of their low toxicity, and due to their ability to target multiple cell signalings, current therapeutic compounds for cancer include phytochemicals [20,21].

Luteolin is one of the most widespread food-derived flavonoids showing preventive and therapeutic effects against several cancer types [22,23,24]. The main purpose of this review is to summarize the most relevant knowledge about the antimetastatic properties of luteolin, emphasizing its interference with key events underlying tumor progression and expansion, namely (i) epithelial-to-mesenchymal transition (EMT), (ii) angiogenesis, (iii) degradation of the extracellular matrix (ECM) and iv) induction of apoptosis (Figure 1). In detail, we report and discuss the most relevant findings from in vitro and in vivo studies (including animal models) that describe the effects of luteolin at both the cellular and molecular level, focusing on major signaling pathways involved in tumor metastasis (Table 1).

## 2. Luteolin

Luteolin is a flavonoid belonging to the flavone family; it is isolated from several vegetables and edible herbs, including radicchio, broccoli, raw brussels sprouts, onion leaves, parsley, carrots, peppers and rosemary, where it can occur either as aglycone or bound to one or several carbohydrates as glycoside [67] (Figure 2).

Luteolin is extracted as a yellow crystalline compound and its chemical structure presents a classic flavone C6-C3-C6 skeleton, consisting in two benzene rings with 4 hydroxyl groups located at positions 3, 4, 5 and 7, and one oxygen-containing ring which presents a C2-C3 double bond [67]. All these functional groups account for the biological/biochemical properties of luteolin, some of them being specifically involved in maintaining redox balance in various pathological processes [68,69] (Figure 3).

In fact, preclinical studies ascribed to luteolin several pharmacological properties, including anti-inflammatory, neuroprotective, antimicrobial/antiviral, cardioprotective, antidiabetic and pro-/antioxidant effects [70]. Interestingly, since this flavone can interact with various signaling pathways, experimental evidence attributes to luteolin important chemopreventive effects, indicating its ability to interfere with almost all cellular processes underlying cancer development, including metastasis formation [23,24,71]. The available data related to the pharmacokinetics of the free or glycosylated forms of luteolin derive mainly from studies on rat models [72,73,74]. Despite the numerous beneficial effects already mentioned, studies on absorption, metabolism and bioavailability of luteolin in humans are very difficult, partly due to its hydrophobicity, which affects bioavailability and limits the yields of the bioactive flavonoid [70].

## 3. Luteolin Affects the Epithelial-Mesenchymal Transition

EMT is the differentiation process of epithelial cells toward mesenchymal ones; several events are observed during this process, including a disorganization of epithelial cell polarity, the dissolution of cellular junctions, as well as a reorganization of the cytoskeleton [75,76,77]. Thus, EMT involves the downregulation of epithelial markers expression, such as E-cadherin, claudins, zonula occludens-1 (ZO-1), and the acquisition of many facets of the mesenchymal cells, including the expression of N-cadherin and vimentin, coupled with a high propensity to cell motility, invasiveness, and resistance to anoikis [76,77,78]. Several signaling pathways have been identified in EMT induction, such as the transforming growth factor-beta (TGF-β), Notch, Wnt/β-catenin, as well as the Hippo-YAP/TAZ pathways [79,80]. Moreover, several transcription factors (and their target genes) promote EMT, including Snail1/2, Twist1/2, ZEB1/2 and hypoxia-inducible factors 1/2 (HIF1/2) [81]. Physiologically, EMT occurs during embryonic development and in tissue remodeling. In cancer, this process assumes a crucial role in promoting metastasis [76,77]. Therefore, blocking or reversing EMT may represent an attractive approach to prevent cancer spreading [77].

As aforementioned, luteolin inhibits or prevents cell invasion and metastasis in several cancer types, due to a modulation of EMT [82]. In this regard, both in vitro and in vivo studies demonstrated that treatment of highly metastatic triple-negative breast cancer (TNBC) with luteolin reduced β-catenin expression and downregulated mesenchymal markers, such as N-cadherin, vimentin, Snail and Slug [25]. Moreover, cancer cells regained their epithelial features by overexpressing cell-cell junctional proteins, such as E-cadherin and claudins [25].

MicroRNAs (miRs) are small, endogenous, noncoding RNAs that can post-transcriptionally regulate gene expression and play an important role in maintaining normal cellular functions [83]. On the other hand, growing evidence shows that some miRs participate in the initiation and progression of cancer, taking part in the EMT process [84,85]. Luteolin was able to reverse EMT in breast cancer cells (MDA-MB-453 and MCF-7) by overexpressing miR-230, which, in turn, inhibited the Ras/Raf/MEK/ERK signaling pathway, known as a marker of cancer invasiveness [26]. Furthermore, luteolin administration significantly inhibited gastric carcinoma (GC) upregulating miR-139, miR-34a, miR-422a, miR-107 levels, while suppressing the oncogenic expression of miR-21, miR-155, miR-224, miR-340 [36]. Regulation of this panel of miRs was accompanied by reduced cell proliferation, cell cycle arrest, and induction of apoptosis [36]. In a different study, the flavonoid inhibited the oncogenic properties of YAP/TAZ in highly metastatic breast cancer, both in vitro and in xenograft models [27]. Moreover, Zang et al. demonstrated that, by interfering with Notch1 and Akt/β-catenin signaling in GC, luteolin reversed the EMT process and, consequently, inhibited tumor progression and invasion, both in vitro and in vivo [35]. By using cultures of human lung adenocarcinoma cells (A549), Chen et al. showed that luteolin inhibited cell proliferation and migration through an attenuation of TGF-β1-induced EMT, by activating the PI3K/Akt–NF-κB–Snail signaling pathway [41]. Furthermore, luteolin (in a time- and dose-dependent manner) reversed IL-6-induced EMT acting on STAT3 signaling and, consequently, reduced the invasiveness of cancer cells by inhibiting the release of metalloproteases (MMPs) in in vitro models of human pancreatic cancer (i.e., Panc-1 and SW1990 cells) [47].

As abovementioned, HIF-1 and HIF-2 are implicated in cancer-associated EMT [81]. Based on these premises, Li et al. observed that luteolin, by inhibiting the HIF-1α/VEGF signaling pathway, decreased the expression of N-cadherin and vimentin mesenchymal markers, while augmenting the level of epithelial cadherin isoform in human and murine melanoma cells (i.e., A375 and B16-F10 cell lines, respectively) [50]. The inversion of hypoxia-induced EMT was also observed in a murine melanoma model, where luteolin inhibited lung metastasis formation by stopping the β3 integrin/FAK signaling pathway [51]. Similarly, luteolin prevented hypoxia-induced EMT of human non-small cell lung carcinoma cells (NSCLC; A549 and NCI-H1975 cell lines), as evidenced by a downregulation of mesenchymal specific markers, such as vimentin and N-cadherin; contextually, luteolin treatment repressed cancer cell motility and adhesion, interfering with integrin β1 expression and with the FAK-signaling pathway [42].

The ability of luteolin to reverse EMT in cancer cells is often enhanced by adding other flavonoids, such as quercetin [86,87]. Indeed, a mixture of luteolin and quercetin was able to attenuate (in a time- and dose-dependent manner) EMT-correlated events, migration and invasiveness of human squamous carcinoma, both in vivo and in vitro, by suppressing the Src/Stat3/S100A signaling pathway [86]. Similarly, the addition of luteolin to quercetin inhibited metastasis of skin squamous carcinoma by blocking the Akt/mTOR/c-Myc signaling pathway to suppress RPS19-activated EMT signaling [87].

In summary, luteolin appears capable to block or reverse EMT by acting on multiple molecular targets, hindering the first steps of cancer spreading (Figure 4).

## 4. Luteolin Suppresses Angiogenesis

Angiogenesis is the process by which new blood vessels form, starting from preexisting vasculature; this event depends on pro-angiogenic mediators, such as vascular endothelial growth factor (VEGF), basic-fibroblast growth factor (bFGF), metalloproteases, etc., and on negative regulators of angiogenesis, including thrombospondin and endostatin [88]. Angiogenesis plays a crucial role during a variety of physiological processes, such as wound healing, embryonic development and pregnancy [88]. On the other hand, neovascularization is a key event for pathological processes, such as tumor progression, invasion and metastatic cascade [89]. This last aspect is pharmacologically relevant: in fact, many authorized cancer therapies are directed against the tumor-associated vessels [89,90,91,92].

Several investigations demonstrate that various flavonoids—including luteolin—act as negative regulators of VEGF and other pro-angiogenic factors [93,94,95]. For instance, Cai and co-workers demonstrated that luteolin decreased VEGF secretion and *VEGF* mRNA expression in pancreatic carcinoma cells (PANC-1, CoLo-357 and BxPC-3 cell lines), via inhibition of the NF-κB transcriptional factor activity [48]. Furthermore, treating human choroidal melanoma cells (C918 and OCM-1) with luteolin not only inhibited VEGF expression, but also increased cancer cell death [54]. Additionally, an in vitro study on breast cancer showed that luteolin, by interfering with Notch1 expression, inhibited VEGF secretion from tumor cells, decreased endothelial cell migration, proliferation, and their propension to form tube-like structures on a Matrigel layer [37]. Cook and co-coworkers demonstrated the ability of luteolin to block the production of VEGF in human breast cancer cells (T47-D and BT-474) responsive to (natural and synthetic) progestins, both in vitro and in a xenograft model [28]. In a similar study, the authors showed that luteolin (in a dose- and time-dependent fashion) significantly reduced VEGF secretion in human TNBC cells (i.e., MDA- MB-435 and highly aggressive MDA-MB-231 (4175) LM2), coupled with a significantly decreased cell viability and reduced migration and invasion in vitro; at the same time, the authors showed that the flavonoid inhibited lung metastasis formation in a dedicated xenograft model [29].

The anti-proliferative/anti-mitotic effect on endothelial cells was also tested in infantile haemangioma in vitro and in vivo, by using haemangioma-derived stem cells (HemSC) [58]. In detail, the results of this study demonstrated that luteolin suppressed VEGF expression and inhibited HemSC growth in a dose-dependent manner, and, at the same time, the flavonoid inhibited both angiogenesis and vasculogenesis, in a murine model, by acting on the frizzled6 (FZD6) signaling pathway [58]. In prostate cancer, luteolin strongly suppressed neovascularization in different experimental settings by inhibiting the activation of VEGFR-2/AKT/ERK/mTOR/P70S6K signaling pathways [56]. In fact, it abolished angiogenesis in an ex vivo chicken chorioallantoic membrane (CAM) assay and in a Matrigel plug assay; in addition, the flavone suppressed both vascularization and growth of the tumor in a xenograft model [56]. Moreover, luteolin inhibited different pathways of vascularization in an in vitro model of uveal melanoma, including angiogenesis, vasculogenesis and vasculogenic mimicry, by suppressing the PI3K/AKT signaling pathway [52].

As previously cited, miRs have a role in cancer evolution, including angiogenesis [85]. In an experimental model of NSCLC, luteolin repressed the expression of the rich element binding protein B (PURB) and crucial proangiogenic factors, including VEGF and MMP-2/-9, by overexpressing the miR-133a-p69 and by regulating MAPK and PI3K/Akt signaling pathways [43]. In breast cancer, the flavonoid upregulated the biogenesis of miR-34a, miR-181a, miR-139-5p, miR-224 and miR-246, while it decreased the level of miR-155, coupled with a significant inhibition of VEGF/Notch signaling and MMPs downregulation [30].

Tumor growth and its progression also depend on proangiogenic factors secreted by the surrounding microenvironment cells (e.g., fibroblasts, tumor associated macrophages (TAMs), etc.) [8]. Fang et al. demonstrated that luteolin inhibited the ability of TAMs to induce angiogenesis, thereby inhibiting tumor growth and its spreading, in both normoxic and hypoxic conditions [96].

Taken together, these studies show promise for luteolin as a potent anti-angiogenic agent, evading tumor evolution and metastatic cascade (Figure 5).

## 5. Luteolin Slows down Extracellular Matrix Degradation

Lysis of extracellular matrix (ECM) molecules occurs through the proteolytic action of a family of zinc-dependent metalloproteases (MMPs), facilitating angiogenesis and promoting cancer cell invasion and dissemination [97,98,99,100]. Luteolin significantly inhibited MMP-2 and MMP-9 (also known as gelatinases) and VEGF expression by suppressing Notch signaling and by modulating specific miRs with a crucial role in breast cancer progression [30]. Another study showed that the flavonoid repressed the invasiveness of androgen receptor-positive TNBC cells by epigenetically downregulating MMP-9 expression (due to a hypoacetylation of the histone H3) via the induction of the AKT/mTOR signaling pathway [31]. A substantial downregulation of migratory propensity was also reported in an in vitro model of human glioblastoma, where an unbalanced MMPs/TIMPs ratio was attributed to inhibition of the pro-invasive p-IGF-1R/PI3K/AKT/mTOR signaling pathway [59]. Furthermore, the flavonoid abolished MMP-2 and MMP-9 activity in a concentration-dependent effect by suppressing pro-invasive Raf/PI3K signaling in murine colorectal carcinoma (CRC) [64]. Recent studies have further reinforced the negative effect of luteolin in CRC dissemination [65]. In fact, in in vitro and in vivo models of this tumor, luteolin induced a downregulation of gelatinases and MMP -3 and -16 expression, coupled with enhanced miR-384 biogenesis, as well as suppression of pleiotrophin (PTN), a small cytokine with a polyhedral role in tumor evolution (PTN promotes cancer cell migration and invasion, and stimulates angiogenesis) [65]. An independent research showed that the flavonoid inhibited the spreading of GC, both in vitro and in a xenograft model, due to a downregulation of MMP-9 associated to a suppression of cMet/Akt/ERK signaling [38]. Chemopreventive properties of the abovementioned compound were further supported by a B16F10 mouse xenograft model, in which the levels of pro-metastatic markers, including MMP-9, MMP-2 and CXCR4, were significantly decreased in the lung tissues isolated from tumor-bearing nude mice after luteolin treatment [53]. Additionally, Shi and co-workers showed that luteolin (in a dose-dependent manner) suppressed the proliferation, migration and invasion of human choroidal melanoma cells in vitro by blocking the secretion of the gelatinases by cancer cells, essentially interfering with the PI3K/Akt signaling pathway [55].

In conclusion, luteolin is able to interfere with different initial steps of the metastatic cascade, including ECM degradation, angiogenesis and cell invasion, by blocking activation pathways of MMPs (Figure 6).

## 6. Luteolin Induces Apoptosis

Apoptosis, a programmed cell death, is an essential mechanism involved in several physiological conditions, including organ development and tissue homeostasis, due to its role in controlling cell depletion, somehow opposing to mitosis in the regulation of cell proliferation [101]. In the tumor context, the intricate machinery of apoptosis plays a polyhedral role, ranging from malignant transformation of cells to metastatic cascade or overstepping anticancer drugs’ resistance, thus suggesting that the balance of pro-survival and pro-death pathways can be impaired at several steps of the apoptotic process [102]. Apoptosis can occur through either the death receptor-mediated extrinsic pathway and/or the mitochondria-mediated intrinsic route, both converging to caspases (cysteine-aspartate proteases) activation, resulting in characteristic morphological and biochemical cellular changes [103].

Apoptotic cellular death can be triggered by different cellular events, including a perturbation of the intracellular redox status, followed by reactive oxygen species (ROS) overproduction and enhanced cellular oxidative stress [44,104]. In this regard, luteolin exhibits pro-apoptotic effects on various cancer cells, acting both on antioxidant activity and on ROS overproduction [45,61,104,105]. Luteolin- induced accumulation of ROS, suppressed NF-κB and potentiated JNK in an in vitro model of NCSLC, hence sensitizing lung cancer cells to TNF-triggered apoptosis [45]. Furthermore, luteolin caused an overproduction of ROS, coupled with the release of cytochrome c from collapsed mitochondria, and with the activation of caspase-3 and -9 in cholangiocarcinoma cells [61]. These molecular changes were associated with inhibition of the Nrf2 transcription factor, with resulting downregulation of its antioxidant target genes, including those encoding γ-glutamylcysteine ligase and heme oxygenase-1 [61]. Kang and co-workers showed that in vitro treatment of human colon cancer (CC) cells (HT-29 cell line) with luteolin was able to cause apoptosis [105]. The authors described an upregulation of intracellular and mitochondrial ROS and enhanced Bax/Bcl-2 ratio, followed by release of cytochrome c from mitochondria to the cytosol and activation of caspase-9 and -3 [105]. Other studies further support the pro-apoptotic effects of the flavone in cancer; in fact, HT-29 cells treated with 60 μM of luteolin for 48 h showed an increased activation of caspase-3, -7, and -9 and Poly-ADP ribose polymerase (PARP) cleavage, followed by the upregulation of several members of the Bcl-2 protein family (Bcl-xL, Mcl-1, survivin, p21 and Mdm-2) [62]. Similarly, luteolin induced apoptosis of human colon tumor cells (HCT-15 cell line), increasing the Bax/Bcl-2 ratio and activating caspase-3 via Wnt/β-catenin/GSK-3β signaling [63]. The intrinsic pro-apoptotic effect of luteolin was also tested in an in vitro model of human GC [39]. In detail, upon exposure to high concentration of the flavonoid (60 µM), GC cells (BGC-823 cell line) showed a significant increase of the Bax/Bcl-2 ratio, coupled with the release of cytochrome c from mitochondria and consequent caspases activation [39]. These molecular events were correlated with the suppression of PI3K/MAPK signaling pathways [39]. In a very extensive study, luteolin treatment induced apoptosis in HeLa cells, by either the intrinsic or extrinsic pathways [106]. Indeed, increased expression of various pro-apoptotic factors, including BAX, BAD, BID, APAF1, TRADD, FAS, FADD, Caspase-3 and -9, correlated with a reduction of anti-apoptotic mediators, such as MCL-1 and BCL-2. Additionally, in this experimental context, luteolin was found to suppress the AKT/mTOR and MAPK/ERK1/2 pathways, highlighting the pro-apoptotic features of protein p53 [106]. Interestingly, luteolin triggered apoptosis through the downregulation of the human telomerase reverse transcriptase (hTERT) via the NF-κB/c-Myc pathway, which correlated to increased Bax/Bcl-2 ratio and caspase-3 protein expression in human breast cancer cells [32]. Another study, performed in HL-60 leukemia cells, demonstrated that luteolin caused a hyperacetylation of histone H3 by activating the ERK and JNKs pathways, with consequential increase of Fas and FasL expression, followed by caspase-8 and -3 activation [66].

In addition to the modulation of transcription events and/or epigenetic changes, several miRs are integrated into critical pathways of cancer spreading [107]. Accordingly, in vitro studies by Han and collaborators demonstrated that luteolin induced apoptosis in prostate cancer (PC3 and LNCaP cell lines) through downregulation of miR-301, coupled with the expression of pro-death DNA-binding effector domain-containing protein 2 (DEDD2) [57]. Another study showed that administration of the flavone to human breast cancer cells (MCF-7) modulated the level of miR-21 and miR-16, reducing cell viability in a dose- and time-dependent manner, but also inducing apoptosis through the intrinsic and extrinsic pathways via upregulation of Bax/Bcl-2 expression [33]. Furthermore, luteolin decreased Bcl-2 expression by upregulating miR-34a in human GC (BGC-823 and SGC-7901 cell lines) [40]. Moreover, Jiang and co-workers demonstrated that luteolin triggered intrinsic apoptosis in NSCLC, in both in vitro and in vivo studies, by enhancing the level of miR-34a-5p and targeting the oncogene MDM4, thus indirectly increasing the availability of the pro-apoptotic features of p53 and p21 [46].

Several cancers acquire drug resistance through different mechanisms, including involvement of PI3K/Akt pathway and tyrosine-kinases activation [4,108,109,110]. Thus, inhibition of PI3K/Akt pathway seems to be crucial to stop cancer progression. Indeed, Lin and co-workers showed that luteolin suppressed PI3K/Akt phosphorylation in an in vitro model of human breast cancer (MCF-7 cells), through the induction of forkhead box O3 (FOXO3a), followed by DNA damage; all these events culminated in mitochondrial apoptotic cascade [34]. The epidermal growth factor receptor (EGFR) is a tyrosine kinase receptor which is commonly upregulated in several types of carcinoma [111]. Various tyrosine kinases inhibitors (TKIs) can suppress EGFR phosphorylation and, consequently, suppress its downstream signaling pathways, including ERK and AKT, with the final effect of inducing apoptosis [112]. Nevertheless, first responses to TKIs (or even to EGFR antibodies) are followed by acquisition of drug resistance for several cancer types. Therefore, studies are consistently searching for new therapeutic alternatives [4,110]. In this scenario, medicinal plants extracts, including luteolin, have been ascribed surprising efficacy as promising anticancer and anti-metastatic compounds [5]. For example, human ductal pancreatic cancer cells (MiaPaCa-2) treated with luteolin showed a markedly decreased activation of EGFR and its downstream protein kinases activity; indeed, treated cells exhibited typical apoptosis features, such as shrinkage of the cell morphology, PARP cleavage and DNA fragmentation [49]. Similarly, luteolin attenuated cell proliferation and growth by inhibiting EGFR/MAP/ERK phosphorylation in an in vitro model of human glioblastoma (U-87 MG and U-251 MG cell lines), inducing apoptosis via PARP cleavage and activation of caspase cascade [60].

To sum up, the anti-metastatic effect of luteolin may depend on its ability to stimulate both intrinsic and extrinsic pathways of apoptosis, by promoting an increase of Bax/Bcl-2 ratio, cytochrome c release and the activation of caspases cascade, by modulating the expression of oncogenic miRs and, in some cases, by overstepping drug resistance (Figure 7).

## 7. Conclusions and Perspectives

Luteolin is able to quench the initial steps of the metastatic cascade. In fact, it inhibits EMT, tumor-associated angiogenesis and ECM degradation. Furthermore, by targeting multiple cellular signals, this flavone induces apoptosis in malignant cells through both the intrinsic pathway and receptor-mediated extrinsic route, with minimal side effects and insignificant toxicity on normal cells [71]. Additionally, luteolin is not only able to overstep drug resistance, but also represents a good adjuvant to other flavonoids, including quercetin [86,87]. However, most of the known activities of luteolin have been investigated only in vitro and/or in animal models, except for a very few non-oncology clinical studies in humans [113,114,115,116,117,118]. Besides, to date, no clinical trials using luteolin as cancer therapy have been performed yet. The main challenges faced by investigators to test this compound in cancer studies include limited funding, absence of product consistency, contamination, and manufacturing difficulties. Indeed, its biological applications are currently limited, due to its hydrophobic nature, although improvement of luteolin bioavailability could be achieved by using nanostructured lipid carriers, microemulsions, and other similar devices [118,119,120,121]. Since preclinical experiments strongly advise for the potential efficacy of luteolin in cancer treatment, the development of strategies that improve its bioavailability, increase its efficacy and decrease its toxicity, will support future clinical studies.

## Figures and Tables

**Figure 1 ijms-24-08824-f001:**
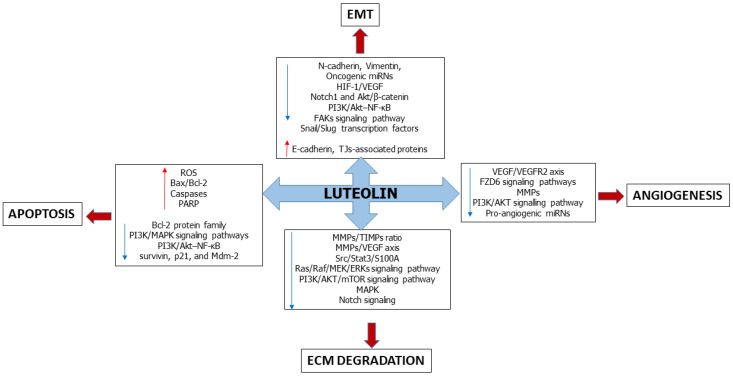
Schematic diagram showing the main molecular targets of luteolin during the initial steps of the metastatic cascade. The flavone interferes with the epithelial-mesenchymal transition (EMT) by inhibiting the expression of N-cadherin, vimentin and several molecular regulators of mesenchymal cell phenotype (e.g., miRNs, HIF-1/VEGF axis, etc.), whereas it enhances the epithelial features. Luteolin exerts anti-angiogenic effects, mainly by blocking the VEGF/VEGFRs signaling pathway and by inhibiting MMPs. Moreover, this compound hampers cell invasiveness by slowing down the extracellular matrix (ECM) degradation through the blockade of MMPs activation and a set of related intracellular pathways. Luteolin induces apoptosis (both the intrinsic and extrinsic routes) by acting on the expression and activities of correlated effectors.

**Figure 2 ijms-24-08824-f002:**
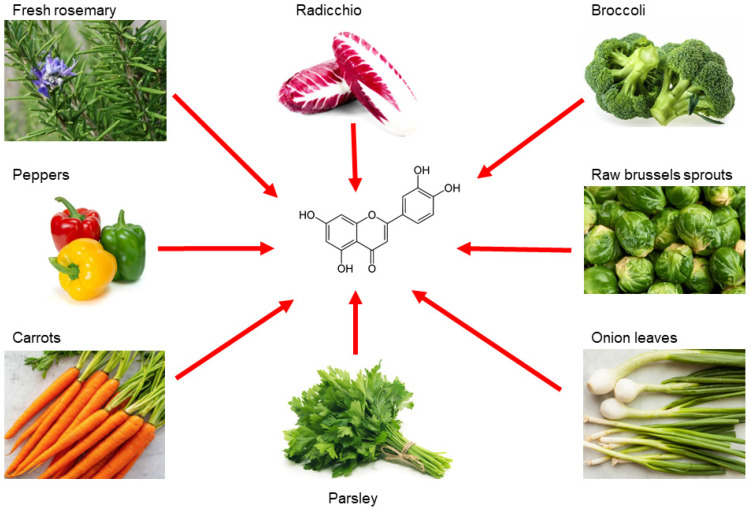
Main sources of luteolin. Edible vegetables, such as radicchio, broccoli, raw brussels sprouts, onion leaves, parsley, carrots, peppers and rosemary are rich in luteolin.

**Figure 3 ijms-24-08824-f003:**
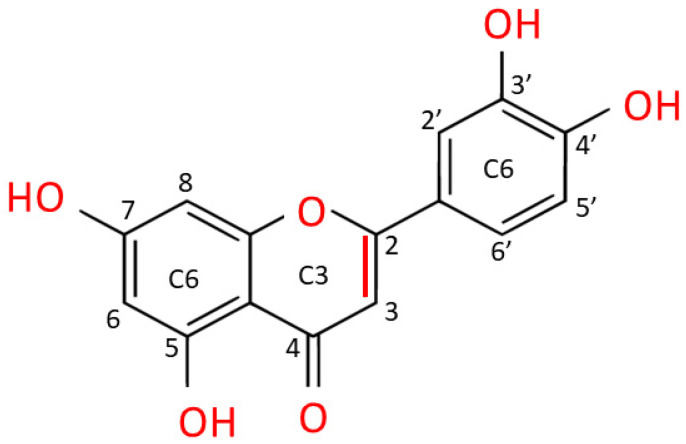
Chemical structure of luteolin (3,4,5,7-tetrahydroxy flavone). Luteolin is a flavone belonging to a group of hydrophobic, naturally occurring compounds, named flavonoids. The functional groups involved in oxido-reductive properties are marked in red.

**Figure 4 ijms-24-08824-f004:**
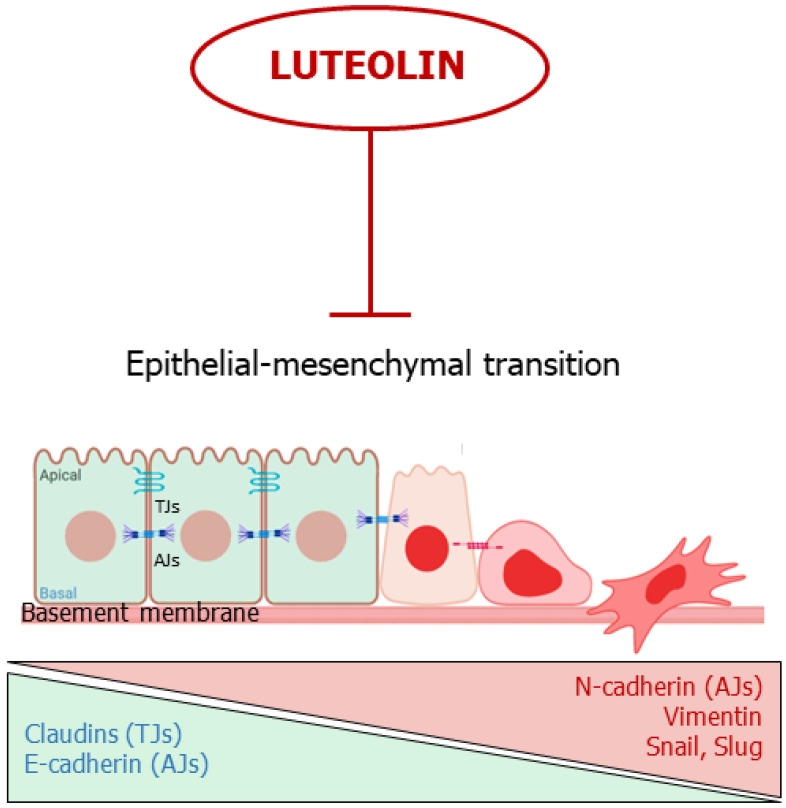
Luteolin interferes with the epithelial-mesenchymal transition. The flavone inhibits or reverses EMT. TJs: Tights Junctions; AJs; Adherens Junctions.

**Figure 5 ijms-24-08824-f005:**
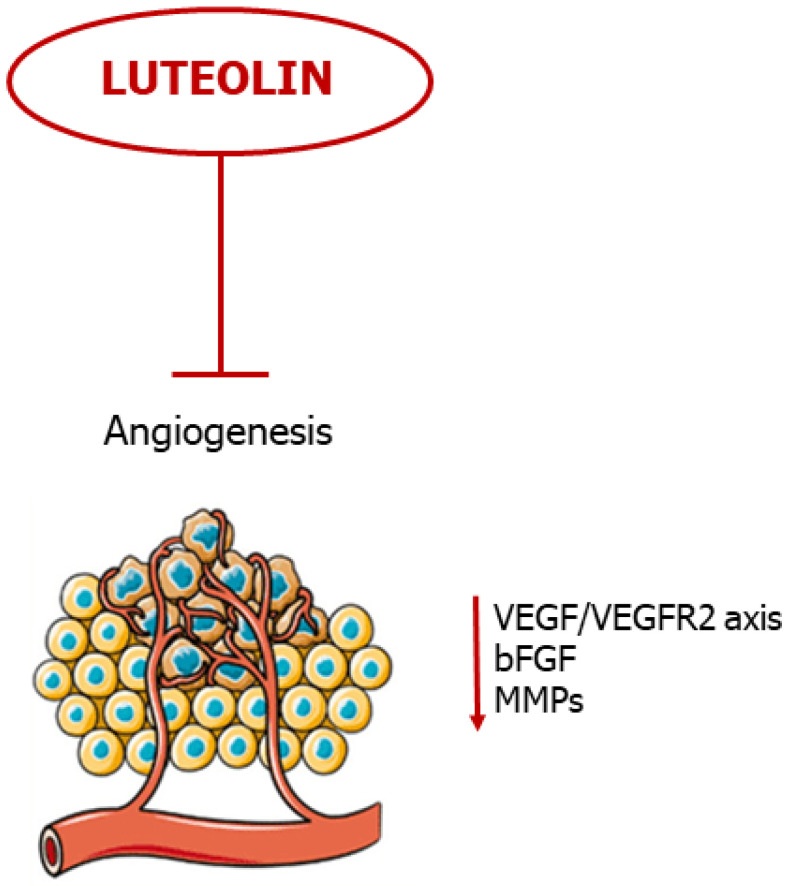
Luteolin blocks neovascularization. The flavonoid blocks angiogenesis by negative regulation of pro-angiogenic mediators.

**Figure 6 ijms-24-08824-f006:**
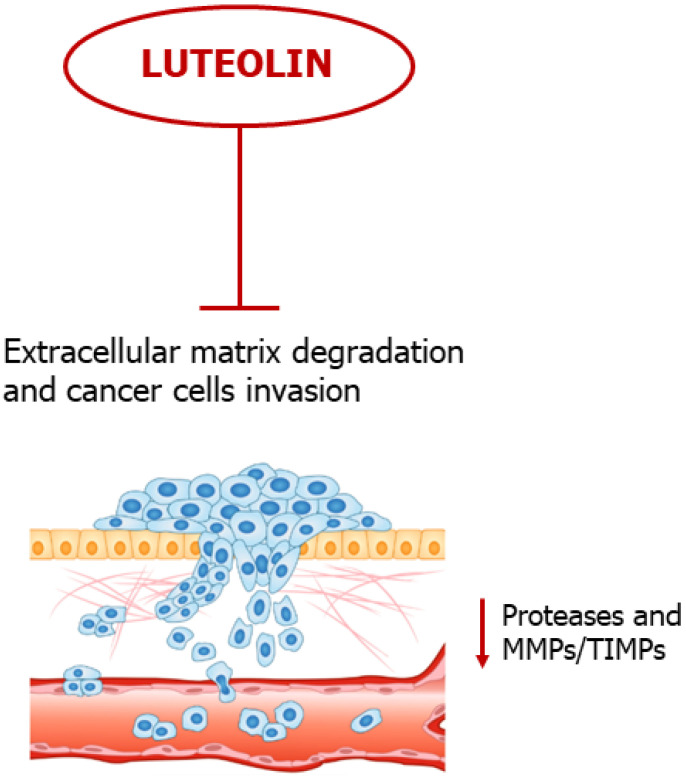
Luteolin slows down ECM lysis. Cancer cell invasion is inhibited by luteolin through suppression of proteases activity.

**Figure 7 ijms-24-08824-f007:**
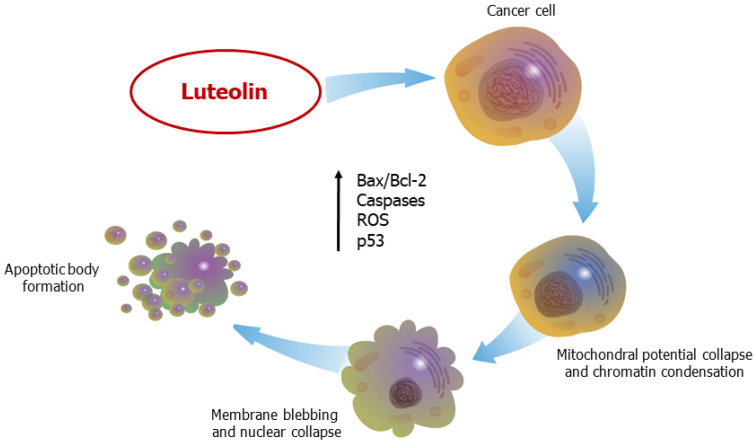
Luteolin induces apoptosis in cancer cells. Both the extrinsic and intrinsic pathways are triggered by luteolin.

**Table 1 ijms-24-08824-t001:** Molecular effects of luteolin in different cancer types.

Tumor Entity	EMT	Angiogenesis	ECM Degradation	Apoptosis
Breast Cancer	Reversal of EMT by suppressing β-catenin signaling; inhibition of cancer cell invasion and metastatic potential [25]. Inhibition of the pro-invasive Ras/Raf/MEK/ERK signaling; increase of miR-203 expression [26]. Increase of E-cadherin expression and reduction of protein levels of fibronectin, N-cadherin and vimentin; decrease of transcriptional activity of YAP/TAZ [27].	Blockade of VEGF secretion within breast cancer cells responsive to natural and synthetic progestins, both in vitro and in a xenograft model [28]. Repression of VEGF secretion by TNBC cells and suppression of their metastatic potential in vitro and in vivo [29]. Repression of Notch signaling and its downstream targets Notch-1, Hes-1, VEGF and gelatinases by regulating the level of oncogenic miRs [30].	Modulation of the biogenesis of specific miRs, inhibition of gelatinases secretion and VEGF/Notch signaling pathway [30]. Epigenetically downregulation of gelatinases expression and activation of AKT/mTOR signaling pathway [31].	Reduction of telomerase expression by targeting NF-κB/c-Myc; increase of Bax/Bcl-2 ratio and caspase-3 [32]. By modulating miR-21 and miR-16, upregulation of Bax/Bcl-2 ratio; triggering of both the intrinsic and the extrinsic pathways of apoptosis [33]. Repression of PI3K/Akt pathway; induction of FOXO3a expression and increase of p21 and p27; induction of PARP cleavage and release of cytochrome *c* [34].
Gastric Carcinoma	Reversal of EMT by inhibiting Akt/β-catenin and Notch signaling pathways [35]. Decreased migration and invasion by regulating Notch1/ PI3K/ AKT/ mTOR/ERK/STAT3 and P38 signaling pathways; regulation of several oncogenic miRs expression in vitro and in vivo [36].	Suppression of VEGF secretion by acting on Notch1 expression and inhibition of the formation of tube-like structures of HUVECs seeded in a Matrigel layer [37].	, Reduction of gelatinases expression via inhibition of cMet/Akt/ERK signaling [38].	Suppression of PI3K/MAPK signaling with increase of Bax/Bcl-2 ratio and cytochrome c release [39]. Decrease of Bcl-2 expression through upregulation of miR-34a [40].
Lung Cancer	Reversal of TGF-β1 induced-EMT by slowing down the activation of PI3K/Akt/IκBa/NF-κB/Snail pathway [41]. Inhibition of hypoxia-induced EMT by blocking integrin β1 expression and FAK-signaling pathway [42].	Repression of VEGF and gelatinases by upregulating miR-133a-p69, and by regulating MAPK/PI3K/Akt signaling pathways [43].	Downregulation of the pro-metastatic markers CXCR4, gelatinases in vitro and in vivo [44].	Induction of ROS accumulation via suppression of SOD activity; suppression of NF-kB potentiating JNK to sensitize cancer cells to TNF [45]. Upregulation of miR-34a-5p via targeting MDM4 oncogene, increase of p53 and p21expression; increase of Bax/Bcl-2 ratio, followed by activation of caspase-3 and -9 [46].
Pancreatic cancer	Deactivation of STAT3 signaling with consequent reversal of Il-6-induced EMT [47].	Decrease of VEGF secretion and VEGF mRNA expression via NF-κB inhibition [48].	-	Attenuation of EGFR signaling pathway and induction of PARP degradation followed by DNA fragmentation [49].
Melanoma	Upregulation of E-cadherin/N-cadherin ratio through inhibition of HIF-1α/VEGF axis [50]. Enhancement of E-cadherin expression via inhibition of β3 integrin/FAK signal pathway in vitro and in vivo [51].	Inhibition of different pathways of tumor neovascularization, including angiogenesis, vasculogenesis and vasculogenic mimicry, by suppressing PI3K/AKT signaling pathway [52].	Downregulation of the pro-metastatic markers, gelatinases and CXCR4 [53].	-
Choroidal melanoma	-	Reduction of VEGF secretion, in a concentration-dependent manner, followed by induction of cell death [54].	Decrease of gelatinase secretion in vitro via inhibition of PI3K/Akt signaling pathway [55].	Increase of Bax/Bcl-2 ratio [52].
Prostate Cancer	-	Inhibition of VEGFR2/AKT/ERK/ mTOR/P70S6K signaling pathway and of neovascularization in ex vivo chicken chorioallantoic membrane (CAM) assay, and in a Matrigel plug assay, as well as in a xenograft model [56].	-	Downregulation of miR-301 that promotes the expression of pro-death DNA-binding effector domain-containing protein 2 (DEDD2) [57].
Haemangioma	-	Suppression of VEGF-A expression with consequential inhibition of microvessel density and vasculogenesis in vivo targeting FZD6 signaling pathway [58].	-	-
Glioblastoma	-	-	Inactivation of the p-IGF-1R/PI3K/AKT/mTOR signaling pathway and alteration of the gelatinase/TIMPs ratio [59].	Stimulation of PARP cleavage, DNA degradation and caspases activation [60].
Colon cancer	-	-	-	Activation of antioxidant enzymes and MAPK signaling; by unbalancing ROS, acting on cytochrome c release and caspase-9 and -3 activation [61]. Activation of caspases 3, 7, 9 and PARP cleavage; downregulation of p21, survivin, Mcl-1, Bcl-x(L) and Mdm-2 [62]. By involving Wnt/β-catenin/GSK-3β signaling, increase of Bax/Bcl-2 ratio and activation of caspase-3 [63].
Colorectal Cancer	-	-	Inactivation of gelatinases by suppressing Raf/PI3K signaling pathways [64]. Downregulation of MMP-2, -9, -3 and -16 expression coupled with an enhancement of miR-384 biogenesis and with suppression of PTN [65].	-
Cholangiocarcinoma	-	-	-	Inhibition of Nrf2 with consequent downregulation of the antioxidant genes γ-glutamylcysteine ligase and heme oxygenase-1, and increase of mitochondrial membrane potential dissipation and caspases -3 and 9 activation [61].
Cervical cancer	-	-	-	Disruption of pro-apoptotic/anti-apoptotic genes equilibrium interfering with the RAS-RAF/MAPK/AKT/ PI3K signaling pathway; triggering collapse of the mitochondrial membrane and DNA fragmentation [33].
Leukemia	-	-	-	Induction of histone H3 hyper-acetylation by activating the ERK /JNKs pathways; increase of Fas and FasL expression culminating in caspases-8/-3 activation [66].

## Data Availability

Not applicable.

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
