# Peer review of "Multi-Faceted Role of Luteolin in Cancer Metastasis: EMT, Angiogenesis, ECM Degradation and Apoptosis"

_ijms, 2023, doi:10.3390/ijms24108824_

Round 1
Reviewer 1 Report
The manuscript "Multi-faceted role of luteolin in cancer metastasis: EMT, angiogenesis, ECM degradation, and apoptosis" presents an interesting topic and it has been properly described. However, some points should be improved:
1. Language editing by a native speaker or a professional translation office is required.
2. What is the significance of this review study?
3. A paragraph about luteolin should be included;
4. The authors may use more figures in the manuscript.
5. Why are clinical trials about luteolin not included in this study?
Language editing by a native speaker or a professional translation office is required.
Author Response
Point-by-point response to reviewer 1
We thank the reviewer for his/her valuable suggestions and appreciation.
We provide a revised version of the manuscript (MS); reviewers’ requests were considered; all modifications are highlighted in red or by track-change.
Reviewer 1:
Comments and Suggestions for Authors
The manuscript "Multi-faceted role of luteolin in cancer metastasis: EMT, angiogenesis, ECM degradation, and apoptosis" presents an interesting topic and it has been properly described. However, some points should be improved:
- Language editing by a native speaker or a professional translation office is required.
Response: language editing was performed by co-authors who speak fluent English.
- What is the significance of this review study?
Response: The purpose of this review has been further defined and clarified in the Introduction paragraph of revised MS (lines 48-56 in red).
- A paragraph about luteolin should be included;
Response: We thank the reviewer for this comment. In the revised version of the MS, a new paragraph about luteolin was added including two new figures, namely figure 2 and 3.
- The authors may use more figures in the manuscript.
Response: As suggested by the reviewer, further figures were added to the revised version of MS. Specifically, figure 3 in the previous version is now presented as figure 1; furthermore, we broke up figure 2 replacing it with four different figures (4, 5, 6 and 7)
- Why are clinical trials about luteolin not included in this study?
Response: clinical trials about luteolin were performed only in non oncological diseases. Therefore, we added the relative references (ref. 114-119) in the conclusion of the revised MS. Furthermore, we explained the possible reasons that justify the lack of human studies testing luteolin in cancer (lines 429-431).
Comments on the Quality of English Language
Language editing by a native speaker or a professional translation office is required.
Response: language editing was performed by co-authors who speak fluent English.
Reviewer 2 Report
The study on the multi-faceted role of luteolin by Rocchetti et al arouses keen interest and adds significantly to the existing knowledge. However, there is a certain need to add few previous relevant works, those which are instrumental in this field. The authors should include the following works and highlight the contribution in relevant sections:
· https://doi.org/10.2147%2FBCTT.S124860
· https://doi.org/10.2147%2FBCTT.S144202
· https://doi.org/10.3390/cancers14215373
Luteolin was discovered in the 1800s but is still lacking clinical relevance. The authors should highlight the shortcomings so far in luteolin research and usage and how they can be overcome. It can be a short paragraph in the Conclusion/Introduction section.
It would be great if the authors can change certain writing style in the manuscript which makes it a bit tiresome to read. It is highly recommended that the authors employ a professional copyediting service for better readability. Overall quality of English is fair, and there is ample opportunity for improvements.
Author Response
Point-by-point response to reviewer 2
We thank the reviewer for his/her valuable suggestions and appreciation.
We provide a revised version of the manuscript (MS); reviewers’ requests were considered; all modifications are highlighted in red or by track-change.
Reviewer 2:
Comments and Suggestions for Authors
The study on the multi-faceted role of luteolin by Rocchetti et al arouses keen interest and adds significantly to the existing knowledge. However, there is a certain need to add few previous relevant works, those which are instrumental in this field. The authors should include the following works and highlight the contribution in relevant sections:
- https://doi.org/10.2147%2FBCTT.S124860
- https://doi.org/10.2147%2FBCTT.S144202
- https://doi.org/10.3390/cancers14215373
Response: We thank the reviewer for these suggestions. Accordingly, we added and discussed properly the recommended works https://doi.org/10.2147%2FBCTT.S144202 and https://doi.org/10.3390/cancers14215373 in the introduction (refs 22 and 24), and the work https://doi.org/10.2147%2FBCTT.S124860 in the paragraph 4 “Luteolin suppresses angiogenesis” (ref. 68) as well as in Table 1 (ref. 68).
Luteolin was discovered in the 1800s but is still lacking clinical relevance. The authors should highlight the shortcomings so far in luteolin research and usage and how they can be overcome. It can be a short paragraph in the Conclusion/Introduction section.
Response: We thank the reviewer for this interesting point. According to his/her comment, we explained the possible reasons that justify the lack of human studies testing luteolin in cancer (lines 429-431).
Comments on the Quality of English Language
It would be great if the authors can change certain writing style in the manuscript which makes it a bit tiresome to read. It is highly recommended that the authors employ a professional copyediting service for better readability. Overall quality of English is fair, and there is ample opportunity for improvements.
Response: language editing was performed by co-authors who speak fluent English.